# Selenonium Salt as a Catalyst for Nucleophilic Substitution Reactions in Water: Synthesis of Thiocyanites and Selenocyanates

**DOI:** 10.3390/molecules28073056

**Published:** 2023-03-29

**Authors:** Alix Y. Bastidas Ángel, Philipe Raphael O. Campos, Eduardo E. Alberto

**Affiliations:** Grupo de Síntese e Catálise Orgânica—GSCO, Departamento de Química, Universidade Federal de Minas Gerais—UFMG, Belo Horizonte 31270-901, Brazil

**Keywords:** selenonium salt, chalcogen bond, organocatalysis, thiocyanate, selenocyanate

## Abstract

Organothiocyanates and selenocyanates are valuable compounds, both in terms of functional group interconversion and due to their biological activities. In this contribution, we report the synthesis of a series of these important substances in a mixture of water and dimethyl carbonate (20/1 proportion) using potassium thio- or selenocyanates salts and organic bromides. The key to the effectiveness of the reaction is a chalcogen bond interaction between a selenonium salt catalyst and the organic substrate.

## 1. Introduction

Research on the interaction of organochalcogen compounds with Lewis bases is an emerging field of study. Organochalcogen compounds can interact with electron-rich species due to the presence of regions of positive electrostatic potential on their surface, which are referred to as σ-holes. The non-covalent interaction of the chalcogen electrophilic site with a Lewis base is defined as a chalcogen bond (ChB) [1,2,3,4]. Chalcogenonium salts (chalcogen IV species) are much better chalcogen bond donors compared with chalcogen (II) compounds, due to the depth of their σ-holes and increased Lewis acidity [5,6,7]. In recent years, research has focused on the application of selenonium salts as catalysts in various organic transformations [8,9,10,11,12,13,14,15,16].

Simple organothiocyanates and organoselenocyanates, as depicted in Figure 1, have been reported to display a broad range of biological activities, ranging from chemoprevention [17] and antiproliferative activity against cancer cells [18] to peroxide scavenging [19] and treatment of Chagas disease [20,21,22]. Additionally, these compounds can be easily converted to other functional groups or employed in synthetic transformations [23,24,25,26].

Not surprisingly, the search for methods to prepare these valuable compounds has attracted much attention. They can be prepared, for instance, with the aid of phase transfer catalysts [27,28,29,30,31], in free radical reactions [32,33,34,35], by electrophilic addition to suitable organic substrates [36,37], or in reactions employing ionic liquids as a solvent and the nucleophilic source of chalcogen cyanide [38]. In this contribution, we share our findings on the use of selenonium salts to activate substrates such as benzyl bromides through chalcogen bond interactions, facilitating the displacement reaction with KChCN (Ch = S or Se) using a 20:1 mixture of water and dimethyl carbonate as a solvent. Reactions are fast (10 to 60 min), conducted at room temperature, and deliver products in good to excellent yields.

## 2. Results and Discussion

Very recently, our group reported the cyanation of benzyl bromides and other organic substrates in water catalyzed by an organoselenide [39]. Our results suggested that the selenonium salt produced throughout the reaction was the alkylating agent, mimicking the behavior of the cofactor S-adenosyl-L-methionine (SAM) in methyltransferase enzymes. Encouraged by these findings, we aimed to extend the protocol to the synthesis of valuable organic thiocyanates and selenocyanates. Initially, we evaluated the thiocyanation of benzyl bromide in anhydrous ethanol using 10 mol% of selenides **C1**–**6** as organocatalysts. Selenide **C1** showed minimal activity, only slightly outperforming the control (Table 1, entries 1 and 2). We tested selenides bearing electron-withdrawing **C2**–**3** or electron-donating groups **C4**–**5**, among which the electron-rich catalyst **C5** displayed the best results, producing the desired product twice as fast as the control reaction (entries 1 and 6). In contrast, selenide **C6**, which exhibited the best catalytic activity in the cyanation of benzyl bromide, performed poorly in the thiocyanation reaction (entry 7). Ultimately, we found that the reaction could be efficiently performed without any organoselenium catalyst by using hydrated ethanol as the solvent. A broad range of organic thiocyanates and selenocyanates was synthesized under these conditions (results presented in the Appendix A).

Inspired by recent reports on the use of organochalcogenides, especially organochalcogenium salts, as chalcogen bond activators in organic transformations [8,9,10,11,12,13,14,15,16], we decided to investigate whether selenonium salt **C7** could catalyze the desired reaction. We compared its activity against **C5**, which was the best catalyst tested previously. Additionally, to make the protocol more attractive, we conducted the experiments using water as a solvent. Selenonium salt **C7** showed excellent activity, delivering the product at a 53% yield in only 10 min of reaction time, while reactions without a catalyst or with **C5** produced hardly any product under the same conditions (Figure 1). It was possible to reduce the catalyst load to 5 mol %, but the reaction time had to be increased to achieve a 50% yield of the product.

One major issue with the procedure using only water as a solvent is its reproducibility. Much more reliable results were found using a 20:1 mixture of water and dimethyl carbonate (DMC) as the solvent. DMC and benzyl bromide are denser than water, so they produce an organic substrate-rich phase, facilitating the mass transport process. Nevertheless, DMC is recognized as an environmentally friendly solvent [40]. The addition of 10 mol% of **C7** to a flask containing benzyl bromide, 1.2 equivalents of KSCN in a 20:1 mixture of water and DMC produced much more product after 10 min than the uncatalyzed reaction (Table 2, entries 1 and 2). Increasing the amount of KSCN to 2.0 equivalents proved optimal, as the result obtained did not change even after extending the reaction time to 1 h (Entries 3 and 4). Catalytic activity was also observed when reducing the amount of **C7** to 5.0 and 2.5 mol% (Entries 5 and 6 respectively). However, replacing benzyl bromide with benzyl chloride or iodide did not result in good yields (results not shown).

Evidence for the activation of benzyl bromide by the Lewis acidity of the selenonium salt **C7** through a chalcogen bond interaction (Figure 2a) was observed by ^1^H NMR. A small, but noticeable signal shift of hydrogen atoms (H_a_) adjacent to the selenonium center was detected when benzyl bromide was added to a solution of **C7** in CDCl_3_. Upon the addition of benzyl bromide, the chemical shift of those hydrogens became lower, as expected, and was directly dependent on the amount of Lewis base added (Figure 2b). Conversely, the ^1^H NMR chemical shift observed for the methylene group of benzyl bromide (H_b_) increased by the same magnitude due to the interaction with the selenonium salt. The largest shift was detected when 0.5 equivalents of BnBr were added relative to **C7**. Adding an excess of BnBr resulted in the chemical shift of H_b_ being almost identical to that of the pure compound. Inconclusive results were obtained from the ^77^Se NMR experiments. Although a very small chemical shift was detected, prolonged exposure to benzyl bromide resulted in the decomposition of the selenonium salt.

Finally, we turned our efforts to broaden the variety of substrates used in this transformation (Table 3). Representative bromides were converted into thiocyanates or selenocyanates upon treatment with KSCN or KSeCN in a mixture of water and DMC catalyzed by selenonium salt **C7**. For benzyl bromide and its congeners assembled with electron-withdrawing groups, the corresponding thio- and selenocyanates were prepared with good to excellent yields after 10 min of reaction (entries 1–7). In most cases, the selenocyanates were produced in better yields, and product formation was not drastically affected by the position of the substituent group. On the other hand, for benzyl bromides bearing electron-donating groups, the reaction time had to be extended to 60 min to achieve better results (entries 8–10). Sterically hindered, heteroaromatic, and α-carbonyl bromides were also satisfactorily converted to the desired products (entries 11–13). The only substrate that did not react under these conditions was 1-bromooctane. Even after 24 h of reaction, no product could be detected (entry 14).

## 3. Materials and Methods

### 3.1. General Remarks

All commercial reagents were used as received. Solvents were analytical grade and purified before use. Moisture-sensitive liquids were transferred using a gas-tight syringe through a rubber septum and stored under argon. Nuclear magnetic resonance (NMR) spectra were determined on Bruker DPX-200 and DRX-400 spectrometers. Chemical shifts (δ) are stated in parts per million (ppm) and coupling constants (J) in Hertz (Hz). Tetramethylsilane (TMS) was used as the internal reference standard for ^1^H NMR, and CDCl_3_ for ^13^C NMR. The following abbreviations are used in the description of NMR data: s = singlet, bs = broad singlet, d = doublet, t = triple, q = quartet, m = multiplet.

### 3.2. Synthesis of Catalysts **C1**–**5**

A dry 50 mL two-neck round-bottom flask equipped with a reflux condenser under argon was charged with the corresponding diselenide (5.0 mmol) and THF (15.0 mL) and stirred at room temperature for 5 min. Then, NaBH_4_ (770.0 mg, 20.0 mmol) was added followed by EtOH (3.0 mL). As soon as the reaction color faded away, a solution of 1-bromobutane (648 μL, 6.0 mmol) in THF (5.0 mL) was added dropwise and the reaction mixture was allowed to stir at 50 °C for 15 h. After this, the mixture was extracted with ethyl acetate (3 × 15 mL). The combined organic extracts were washed with water (1 × 20 mL), dried over MgSO_4,_ and solvents were evaporated under reduced pressure.

Butyl(phenyl)selane (**C1**) [41] was purified by silica gel chromatography column (hexanes) to afford a pale-yellow oil, 0.98 g (46% yield). ^1^H NMR (CDCl_3_, 400 MHz) δ = 7.49–7.48 (m, 2H), 7.27–7.21 (m, 2H), 2.91 (t, *J* = 7.4 Hz, 2H), 1,69 (pentet, *J* = 7.4 Hz, 2H), 1.42 (sextet, *J* = 7.4 Hz, 2H), 0.90 (t, *J* = 7.4 Hz, 3H). ^13^C NMR (CDCl_3_, 100 MHz) δ = 132.5, 130.9, 129.1, 126.7, 32.4, 27.7, 23.1, 13.7.

Butyl(4-chlorophenyl)selane (**C2**) [41] was obtained pure after work-up as a pale-yellow oil, 2.3 g (93% yield). ^1^H NMR (CDCl_3_, 400 MHz) δ = 7.39 (d, *J* = 8.5 Hz, 2H), 7.21 (d, *J* = 8.5 Hz, 2H), 2.89 (t, *J* = 7.5 Hz, 2H), 1.66 (pentet, *J* = 7.5 Hz, 2H), 1.41 (sextet, *J* = 7.5 Hz, 2H), 0.90 (t, *J* = 7.4 Hz, 3H). ^13^C NMR (CDCl_3_, 100 MHz) δ = 133.9, 132.9, 129.3, 129.0, 32.3, 28.2, 23.1, 13.7.

(3,5-bis(trifluoromethyl)phenyl)(butyl)selane (**C3**) [41] was purified by silica gel chromatography column (hexanes) to afford a pale-yellow oil, 2.27 g (65% yield). ^1^H NMR (CDCl_3_, 400 MHz) δ = 7.85 (s, 2H), 7.69 (s, 2H), 3.03 (t, *J* = 7.4 Hz, 2H), 1.73 (pentet, *J* = 7.5 Hz, 2H), 1.46 (sextet, *J* = 7.4 Hz, 2H), 0.94 (t, *J* = 7.4 Hz, 3H). ^13^C NMR (CDCl_3_, 100 MHz) δ = 134.5, 132.2 (q, *J* = 32.8 Hz), 131.3 (d, *J* = 3.07 Hz), 123.3 (q, *J* = 271.3 Hz), 120.3 (pentet, *J* = 3.75 Hz), 31.9, 28.0, 23.1, 13.6.

Butyl(p-tolyl)selane (**C4**) [41] was obtained pure after work-up as a colorless oil, 1.5 g (66% yield). ^1^H NMR (CDCl_3_, 400 MHz) δ = 7.38 (d, *J* = 8.0 Hz, 2H), 7.05 (d, *J* = 8.0 Hz, 2H), 2.86 (t, *J* = 7,4 Hz, 2H), 2.30 (s, 3H), 1.66 (pentet, *J* = 7.5 Hz, 2H), 1.41 (sextet, *J* = 7.5 Hz, 2H), 0.89 (t, *J* = 7.4 Hz, 3H). ^13^C NMR (CDCl_3_, 100 MHz) δ = 136.8, 133.1, 129.9, 126.9, 32.5, 28.1, 23.1, 21.2, 13.7.

Butyl(4-methoxyphenyl)selane (**C5**) [42] was purified by silica gel chromatography column (hexanes/AcOEt = 9/1) to afford a pale-yellow oil, 1.3 g (54% yield). ^1^H NMR (CDCl_3_, 400 MHz) δ = 7.45 (d, *J* = 8.7 Hz, 2H), 6.79 (d, *J* = 8.7 Hz, 2H), 3.77 (s, 3H), 2.81 (t, *J* = 7.5 Hz, 2H), 1.63 (pentet, *J* = 7.6 Hz, 2H), 1.39 (sextet, *J* = 7.5 Hz, 2H), 0.88 (t, *J* = 7.4 Hz, 3H). ^13^C NMR (CDCl_3_, 100 MHz) δ = 159.3, 135.6, 120.4, 114.8, 55.4, 32.5, 28.9, 22.9, 13.7.

### 3.3. Synthesis of Catalyst 4-(butylselanyl)benzoic Acid (**C6**)

Prepared as described in reference [39].

### 3.4. Synthesis of Catalyst Dibutyl(phenyl)selenonium Tetrafluoroborate (C7)

A dry 25 mL one-neck round-bottom flask under argon was charged with Butyl(phenyl)selane **C1** (1.07 g, 5.0 mmol) and 1-bromobutane (1.6 mL, 15 mmol). The mixture was stirred until it became homogeneous, and AgBF_4_ (1.07 g, 5.5 mmol) was added. After stirring for 6 h at room temperature in the dark, dichloromethane (3.0 mL) was added. After 5 min the mixture was filtered through a pad of celite, activated charcoal was added to the solution, and then it was filtered again through a pad of celite. The solvent was evaporated under reduced pressure and the product was washed with diethyl ether (3.0 mL). After decantation, the solvent was removed with the aid of a pipette. This step was repeated three times, and then the resulting product was dried in a high vacuum pump. Catalyst **C7** was obtained as a viscous colorless oil, 0.84 g (47% yield) [8]. ^1^H NMR (CDCl_3_, 400 MHz) δ = 7.88–7.86 (m, 2H), 7.74–7.67 (m, 3H), 3.86–3.79 (m, 2H), 3.73–3.66 (m, 2H), 1.79–1.59 (m, 4H), 1.49–1.42 (m, 4H), 0.89 (t, *J* = 7.3 Hz, 6H). ^13^C NMR (CDCl_3_, 100 MHz) δ = 133.8, 131.7, 131.4, 122.2, 43.1, 27.5, 22.4, 13.44. ^77^Se NMR (CDCl_3_, 76 MHz) δ = 399.6.

### 3.5. Synthesis of Thio- and Selenocyanates

A test tube was charged with catalyst **C7** (18 mg, 0.05 mmol), dimethyl carbonate (0. 1 mL), and the corresponding substrate (0.5 mmol). The mixture was stirred until it became homogeneous, and then KSCN (97.2 mg, 1.0 mmol) or KSeCN (144.1 mg, 1.0 mmol) diluted in water (2.0 mL) was added. The reaction mixture was stirred at 25 ± 2 °C (water bath) at a constant rate of 360 rpm for the time indicated in Table 3. Then, the mixture was extracted with AcOEt (3 × 10.0 mL) and the combined organic phases were washed with water (3 × 10.0 mL) and brine (3 × 10.0 mL), dried over MgSO_4,_ and evaporated under reduced pressure. Purification was performed by a silica gel chromatography column with mixtures of hexanes and AcOEt.

(thiocyanatomethyl)benzene (**1a**) [33] was purified by silica gel chromatography column (hexanes/AcOEt = 9/1) to afford a pale-yellow solid, 64.2 mg (86% yield); m.p. = 39.0–40.0 °C; ^1^H NMR (CDCl_3_, 400 MHz) δ = 7.39–7.32 (m, 5H), 4.12 (s, 2H). ^13^C NMR (CDCl_3_, 100 MHz) δ = 134.5, 129.2, 129.1, 128.9, 112.1, 38.4.

(selenocyanatomethyl)benzene (**1b**) [35] was purified by silica gel chromatography column (hexanes/AcOEt = 9/1) to afford a white solid, 92.2 mg (94% yield); m.p. = 67.5–69.0 °C (lit. = 71.0–73.0 °C); ^1^H NMR (CDCl_3_, 400 MHz) δ = 7.37–7.35 (m, 5H), 4.30 (s, 2H). ^13^C NMR (CDCl_3_, 100 MHz) δ = 135.6, 129.3, 129.2, 128.9, 102.0, 32.9.

1-chloro-4-(thiocyanatomethyl)benzene (**2a**) [33] was purified by silica gel chromatography column (hexanes/AcOEt = 9/1) to afford a yellow oil, 79.0 mg (86% yield); ^1^H NMR (CDCl_3_, 400 MHz) δ = 7.36–7.34 (m, 2H), 7.29–7.27 (m, 2H), 4.09 (s, 2H). ^13^C NMR (CDCl_3_, 100 MHz) δ = 130.0, 131.1, 130.4, 129.4, 111.8, 37.6.

1-chloro-4-(selenocyanatomethyl)benzene (**2b**) [35] was purified by silica gel chromatography column (hexanes/AcOEt = 9/1) to afford a yellow solid, 111.8 mg (97% yield); m.p. = 55.0–55.5 °C (lit. = 56.0–58.0 °C); ^1^H NMR (CDCl_3_, 400 MHz) δ = 7.35–7.28 (m, 4H), 4.23 (s, 2H). ^13^C NMR (CDCl_3_, 100 MHz) δ = 134.8, 134.3, 130.5, 129.5, 101.7, 31.9.

1-chloro-2-(thiocyanatomethyl)benzene (**3a**) [33] was purified by silica gel chromatography column (hexanes/AcOEt = 9/1) to afford a yellow oil, 79.0 mg (86% yield); ^1^H NMR (CDCl_3_, 400 MHz) δ = 7.44–7.39 (m, 2H), 7.34–7.27 (m, 2H), 4.23 (s, 2H). ^13^C NMR (CDCl_3_, 100 MHz) δ = 134.3, 132.4, 131.3, 130.6, 130.2, 127.6, 36.3.

1-chloro-2-(selenocyanatomethyl)benzene (**3b**) [43] was purified by silica gel chromatography column (hexanes/AcOEt = 9/1) to afford a yellow oil, 100.3 mg (87% yield); ^1^H NMR (CDCl_3_, 400 MHz) δ = 7.42–7.37 (m, 2H), 7.29–7.25 (m, 2H), 4.31 (s, 2H). ^13^C NMR (CDCl_3_, 100 MHz) δ = 134.1, 133.8, 130.9, 130.3, 130.1, 127.5, 101.8, 30.6.

1-fluoro-4-(thiocyanatomethyl)benzene (**4a**) [44] was purified by silica gel chromatography column (hexanes/AcOEt = 9/1) to afford a yellow oil, 70.2 mg (84% yield); ^1^H NMR (CDCl_3_, 400 MHz) δ = 7.36–7.32 (m, 2H), 7.09–7.05 (m, 2H), 4,12 (s, 2H). ^13^C NMR (CDCl_3_, 100 MHz) δ = 163.0 (d, *J* = 246.9 Hz), 130.9 (d, *J* = 8.6 Hz); 130.4 (d, *J* = 3.4 Hz), 116.3 (d, *J* = 21.9 Hz), 119.9, 37.7.

1-fluoro-4-(selenocyanatomethyl)benzene (**4b**) [45]was purified by silica gel chromatography column (hexanes/AcOEt = 9/1) to afford a white solid, 79.2 mg (74% yield); m.p. = 62.4–62.5 °C (lit. = 64.0–65.0 °C); ^1^H NMR (CDCl_3_, 400 MHz) δ = 7.36–7.32 (m, 2H), 7.07–7.03 (m, 2H), 4.26 (s, 2H). ^13^C NMR (CDCl_3_, 100 MHz) δ = 162.8 (d, *J* = 246.8 Hz), 131.6 (d, *J* = 3.2 Hz), 130.9 (d, *J* = 8.5 Hz), 116.2 (d, *J* = 21.6 Hz), 101.9, 32.0.

1-fluoro-2-(thiocyanatomethyl)benzene (**5a**) [46] was purified by silica gel chromatography column (hexanes/AcOEt = 9/1) to afford a yellow oil, 60.0 mg (61% yield); ^1^H NMR (CDCl_3_, 400 MHz) δ = 7.39–7.33 (m, 2H), 7.19–7.09 (m, 2H), 4.19 (s, 2H). ^13^C NMR (CDCl_3_, 100 MHz) δ = 160.9 (d, *J* = 247.7 Hz), 131.2, 131.1 (d, *J* = 3.1 Hz), 124.9 (d, *J* = 3.7 Hz), 122.1 (d, *J* = 14.5 Hz), 116.1 (d, *J* = 20.7 Hz), 111.8, 31.8 (d, *J* = 3.5 Hz).

1-fluoro-2-(selenocyanatomethyl)benzene (**5b**) [47] was purified by silica gel chromatography column (hexanes/AcOEt = 9/1) to afford a white solid, 87.8 mg (82% yield); m.p. = 60.5–61.5 °C (lit. = 65.0–66.0 °C); ^1^H NMR (CDCl_3_, 400 MHz) δ = 7.38–7.26 (m, 2H), 7.17–7.07 (m, 2H), 4.26 (s, 2H). ^13^C NMR (CDCl_3_, 100 MHz) δ = 160.8 (d, *J* = 247.8 Hz), 131.0 (d, *J* = 3.1 Hz), 130.8 (d, *J* = 8.5 Hz), 124.8 (d, *J* = 3.8 Hz), 127.4 (d, *J* = 14.3 Hz), 115.9 (d, *J* = 20.6 Hz), 101.7, 25.6 (d, *J* = 3.6 Hz).

1-nitro-4-(thiocyanatomethyl)benzene (**6a**) [48] was purified by silica gel chromatography column (hexanes/AcOEt = 95/5) to afford a white solid, 73.8 mg (76% yield); m.p. = 82.2–84.4 °C; ^1^H NMR (CDCl_3_, 400 MHz) δ = 8.25 (d, *J* = 8.7 Hz, 2H), 7.58 (d, *J* = 8.7 Hz, 2H), 4.23 (s, 2H). ^13^C NMR (CDCl_3_, 100 MHz) δ = 148.1, 141.9, 130.1, 124.4, 11.2, 36.9.

1-nitro-4-(selenocyanatomethyl)benzene (**6b**) [35] was purified by silica gel chromatography column (hexanes/AcOEt = 95/5) to afford a yellow solid, 62.7 mg (52% yield); m.p. = 112.9–113.1 °C (lit. = 122.0–124.0 °C); ^1^H NMR (CDCl_3_, 400 MHz) δ = 8.24 (d, *J* = 8.6 Hz, 2H), 7.55 (d, *J* = 8.6 Hz, 2H), 4.31 (s, 2H). ^13^C NMR (CDCl_3_, 100 MHz) δ = 148.0, 143.4, 130.1, 124.5, 100.9, 31.1.

1-nitro-2-(thiocyanatomethyl)benzene (**7a**) [49] was purified by silica gel chromatography column (hexanes/AcOEt = 95/5) to afford a yellow solid, 66.0 mg (68% yield); m.p. = 67.0–68.0 °C (lit. = 69.0–71.0 °C); ^1^H NMR (CDCl_3_, 400 MHz) δ = 8.24–8.22 (m, 1H), 7.74–7.70 (m, 1H), 7.63–7.58 (m, 1H), 7.56–7.54 (m, 1H), 4.46 (s, 2H). ^13^C NMR (CDCl_3_, 100 MHz) δ = 147.1, 134.7, 132.6, 131.2, 130.5, 126.3, 112.2, 36.7.

1-nitro-2-(selenocyanatomethyl)benzene (**7b**) [35] was purified by silica gel chromatography column (hexanes/AcOEt = 95/5) to afford a yellow solid, 98.9 mg (82% yield); m.p. = 74.0–74.5 °C (lit. = 72.0–74.0 °C); ^1^H NMR (CDCl_3_, 400 MHz) δ = 8.23–8.19 (m, 1H), 7.73–7.69 (m, 1H), 7.59–7.54 (m, 2H), 4.46 (s, 2H), ^13^C NMR (CDCl_3_, 100 MHz) δ = 146.6, 134.9, 133.3, 132.1, 130.1, 126.2, 102.8, 30.7.

1-ethyl-4-(thiocyanatomethyl)benzene (**8a**) [50] was purified by silica gel chromatography column (hexanes/AcOEt = 95/5) to afford a yellow oil, 77.1 mg (87% yield); ^1^H NMR (CDCl_3_, 400 MHz) δ = 7.26 (d, *J* = 8.2 Hz, 2H), 7.19 (d, *J* = 8.2 Hz, 2H), 4.11 (s, 2H), 2.64 (q, *J* = 7.6 Hz, 2H), 1.22 (t, *J* = 7.6 Hz, 3H). ^13^C NMR (CDCl_3_, 100 MHz) δ = 145.2, 131.6, 129.1, 128.7, 112.3, 38.4, 28.7, 15.5.

1-ethyl-4-(selenocyanatomethyl)benzene (**8b**) [35] was purified by silica gel chromatography column (hexanes/AcOEt = 95/5) to afford a yellow oil, 77.3 mg (69% yield); ^1^H NMR (CDCl_3_, 400 MHz) δ = 7.27 (d, *J* = 7.6 Hz, 2H), 7.18 (d, *J* = 7.6 Hz, 2H), 4.29 (s, 2H), 2.65 (q, *J* = 7.6 Hz, 2H), 1.23 (t, *J* = 7.6 Hz, 2H). ^13^C NMR (CDCl_3_, 100 MHz) δ = 145.1, 132.6, 129.2, 128.8, 102.3, 33.0, 28.7, 15.5.

1-methoxy-4-(thiocyanatomethyl)benzene (**9a**) [33] was purified by silica gel chromatography column (hexanes/AcOEt = 9/1) to afford a yellow oil, 82.4 mg (92% yield); ^1^H NMR (CDCl_3_, 400 MHz) δ = 7.28–7.25 (m, 2H), 6.90–6.88 (m, 2H), 4.12 (s, 2H), 3.79 (s, 3H). ^13^C NMR (CDCl_3_, 100 MHz) δ = 160.1, 130.4, 126.4, 114.6, 112.3, 55.4, 38.3.

1-methoxy-4-(selenocyanatomethyl)benzene (**9b**) [35] was purified by silica gel chromatography column (hexanes/AcOEt = 9/1) to afford a yellow solid, 90.4 mg (80% yield); m.p. = 53.5–55.0 °C (lit. = 52.0–54.0 °C); ^1^H NMR (CDCl_3_, 400 MHz) δ = 7.29–7.26 (m, 2H), 6.89–6.85 (m, 2H), 4.27 (s, 2H), 3.79 (m, 3H). ^13^C NMR (CDCl_3_, 100 MHz) δ = 159.9, 130.4, 127.4, 114.6, 102.4, 55.4, 32.9.

5-(thiocyanatomethyl)benzo[d][1,3]dioxole (**10a**) [51] was purified by silica gel chromatography column (hexanes/AcOEt = 9/1) to afford a colorless oil, 58.0 mg (60% yield); ^1^H NMR (CDCl_3_, 400 MHz) δ = 6.83–6.77 (m, 3H), 5.97 (s, 2H), 4.09 (s, 2H). ^13^C NMR (CDCl_3_, 100 MHz) δ = 148.3, 127.9, 123.0, 112.2, 109.2, 108.7, 101.6, 38.8.

5-(selenocyanatomethyl)benzo[d][1,3]dioxole (**10b**) [52] was purified by silica gel chromatography column (hexanes/AcOEt = 9/1) to afford a yellow solid, 97.2 mg (81% yield); m.p. = 70.0–71.5 °C; ^1^H NMR (CDCl_3_, 400 MHz) δ = 6.82–6.81 (m, 2H), 6.77–6.75 (m, 1H), 5.96 (s, 2H), 4.23 (s, 2H). ^13^C NMR (CDCl_3_, 100 MHz) δ = 148.2, 148.1, 129.1, 122.9, 109.2, 108.7, 102.2, 101.5, 33.4.

(1-thiocyanatoethyl)benzene (**11a**) [53] was purified by silica gel chromatography column (hexanes/AcOEt = 95/5) to afford a yellow oil, 30.2 mg (37% yield); ^1^H NMR (CDCl_3_, 400 MHz) δ = 7.39–7.32 (m, 5H), 4.60 (q, *J* =7.0 Hz, 1H), 1.87 (d, *J* =7.0, 3H). ^13^C NMR (CDCl_3_, 100 MHz) δ = 139.2, 129.3, 129.2, 127.3, 111.9, 48.7, 22.2.

(1-selenocyanatoethyl)benzene (**11b**) [35] was purified by silica gel chromatography column (hexanes/AcOEt = 95/5) to afford a yellow oil, 43.1 mg (41% yield); ^1^H NMR (CDCl_3_, 400 MHz) δ = 7.38–7.30 (m, 5H), 4.90 (q, *J* = 6,8 Hz, 1H), 2.04 (d, *J* = 6.9, 3H) ^13^C NMR (CDCl_3_, 100 MHz) δ = 139.6, 129.2, 128.9, 127.2, 102.7, 45.7, 22.9.

2-(thiocyanatomethyl)thiophene (**12a**) [33] was purified by silica gel chromatography column (hexanes/AcOEt = 9/1) to afford a yellow oil, 49.7 mg (64% yield); ^1^H NMR (CDCl_3_, 400 MHz) δ = 7.33–7.31 (m, 1H), 7.12–7.11 (m, 1H), 6.99–6.96 (m, 1H), 4.39 (s, 2H). ^13^C NMR (CDCl_3_, 100 MHz) δ = 136.2, 128.8, 127.5, 127.4, 111.8, 33.3.

2-(selenocyanatomethyl)thiophene (**12b**) [54] was purified by silica gel chromatography column (hexanes/AcOEt = 9/1) to afford a yellow solid, 72.8 mg (72% yield); m.p. = 53.0–55.0 °C (lit. = 48.0–50.0 °C); ^1^H NMR (CDCl_3_, 400 MHz) δ = 7.30–7.29 (m, 1H), 7.12- 7.11 (m, 1H), 6.97- 6.95 (m, 1H), 4.54 (s, 2H). ^13^C NMR (CDCl_3_, 100 MHz) δ = 137.6, 128.7, 127.6, 127.2, 102.1, 27.0.

1-(4-bromophenyl)-2-thiocyanatoethan-1-one (**13a**) [55] was purified by silica gel chromatography column (hexanes/AcOEt = 9/1) to afford a white crystalline solid, 76.8 mg (60% yield); m.p. = 145.5–146.6 °C (lit. = 148.8–149.2 °C); ^1^H NMR (CDCl_3_, 400 MHz) δ = 7.80 (d, *J* = 8.6 Hz, 2H), 7.68 (d, *J* = 8.6 Hz, 2H), 4.69 (s, 2H). ^13^C NMR (CDCl_3_, 100 MHz) δ = 190.1, 132.9, 132.8, 130.5, 130.0, 111.7, 42.8.

1-(4-bromophenyl)-2-selenocyanatoethan-1-one (**13b**) [56] was purified by silica gel chromatography column (hexanes/AcOEt = 9/1) to afford a yellow solid, 106.1 mg (70% yield); m.p. = 138.4–138.6 °C (lit. = 144.0–145.0 °C); ^1^H NMR (CDCl_3_, 400 MHz) δ = 7.82 (d, *J* = 8.6 Hz, 2H), 7.68 (d, *J* = 8.6 Hz, 2H), 4.86 (s, 2H). ^13^C NMR (CDCl_3_, 100 MHz) δ = 192.4, 132.8, 132.7, 130.7, 130.3, 101.7, 38.1.

## 4. Conclusions

In this study, we developed a simple and efficient protocol to produce a range of thio- and selenocyanates using a sustainable solvent mixture of water and dimethyl carbonate. The reaction was only possible with the activation of substrates using a catalytic amount of selenonium salt. Our experimental evidence showed that selenonium salts are superior catalysts compared with organoselenides and that activation occurred through chalcogen bond interaction, as demonstrated by ^1^H NMR experiments. We successfully demonstrated the synthesis of thio- and selenocyanates with various electron-withdrawing or electron-donating groups, as well as sterically hindered, heteroaromatic, and α-carbonyl substrates.

## Data Availability

Data are contained within the article and Appendix A.

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
