# Peer review of "Selenonium Salt as a Catalyst for Nucleophilic Substitution Reactions in Water: Synthesis of Thiocyanites and Selenocyanates"

_molecules, 2023, doi:10.3390/molecules28073056_

Round 1

Reviewer 1 Report

The authors present a new procedure to obtain organothiocyanates and organoselenocyanates through the reaction of organic bromides with potassium thio- or selenocyanate salts. A mixture of water and dimethyl carbonate (20/1) was used as the solvent. Chalcogen bonding was presented as key interaction that enables an effective transformation. The material presents an alternative way to obtain simple organothiocyanates and organoselenocyanates using a more sustainable synthetic procedure. I recommend the article for publication, however, a few issues have to be addressed.

1.       As the biological potential of these derivatives is highlighted in the text, a presentation of some examples of bio-active compounds (mentioned in lines 28-30, references 5-8) could be added to Figure 1, besides the general structure of organothiocyanates and organoselenocyanates.

2.       Line 68 – “Inspired by recent reports on the use of organochalcogenides, especially organochalcogenium salts, as Chalcogen Bond activators in organic transformations,” – a reference to these reports should be added.

3.       Did the authors test the reaction using different proportion of DMC-water? 

Author Response

We appreciate the valuable comments and suggestions provided by the reviewer. As suggested, we have revised Figure 1 to include the structures of thiocyanates and selenocyanates with biological activity and included the appropriate reference to the statement on line 68.

Regarding the inquiry about experiments using different proportions of water/DMC, we did not carry out such a test. Although we did not mention the results in our manuscript, we explored this possibility during the optimization of the reaction conditions using another solvent. Our observations revealed that increasing the amount of the organic solvent led to a slower reaction while decreasing the amount of the co-solvent resulted in reproducibility issues.

Reviewer 2 Report

The paper entitled” Selenonium Salt as a Catalyst for Nucleophilic Substitution Reactions in Water: Synthesis of Thiocyanides and Selenocyanates” by A. Y. Bastidas Ángel, P. R. O. Campos and E. E. Alberto reports the synthesis of a series of thio- and seleno-cyanates employing a water/dimethyl carbonate mixture as solvent at room temperature.

The key of such synthesis is the use of a selenonium salt which catalyzes such reactions establishing a chalcogen bond with bromide substrates.

The manuscript deserves the publication on “Molecules”.

I have only minor points that should be addressed by the authors:

          1)  Line 21. The authors introduce the concept of sigma-hole. I suggest to provide some references to link the readers to more insight about such a concept. In particular, I think the following paper can give a more appropriate definition of it (DOI: 10.1039/c7cp06793c).         

          2) Can be deduced a correlation between the type of substituent and the thio-/seleno-derivate products with the obtained yields in terms of the corresponding strength of the chalcogen bond formed?

Author Response

We want to thank the reviewer for the suggestions and comments. We've added two more references related to the "sigma-hole" concept, as suggested.

Unfortunately, we can't properly answer the question about whether the strength of ChB correlates with how well reactions work with nucleophiles. If on one hand, substrates assembled with EWDG would, in theory, make it harder for ChB to form with the catalyst. On the other hand, substrates with EDG are inherently less reactive to nucleophilic displacements. So, we need to do more research on this to figure it out. We're thinking theoretical calculations might help us determine how "effective" ChB is in different substrates.

Reviewer 3 Report

This paper presents the synthesis of thiocyanates and selenocyanates by the reaction of the corresponding benzyl-type bromides with KChCN (Ch = S, Se) in the presence of the selenonium salt [Ph(Bu)2Se]+BF4- as the catalyst, which produces the desired compounds in high-to-moderate yields except octyl-ChCN. This is a convenient and good procedure to be published in Molecules, however, a weak point of this paper is on the chalcogen bond interaction between [Ph(Bu)2Se]+BF4- and PhCH2Br studied with 1H NMR; the high-field shift of CH2Se+ protons in the 1H NMR experiment is only 0.03 ppm. The authors should carefully check the shift of benzyl protons of PhCH2Br and should observe the change of the  77Se NMR chemical shift that would be directly influenced by the chalcogen bond. If possible, theoretical consideration may be added for the interaction. This reviewer thinks that it is important for increasing the value of this study to distinguish the chalcogen bond interaction from a simple role as a phase-transfer catalyst.

Author Response

We appreciate the insight and suggestions provided by the reviewer. We have incorporated the data and discussion on 1H NMR chemical shifts observed for the methylene group of benzyl bromide when it interacts with the selenonium salt in the final version of the manuscript. This data reinforces the postulated formation of ChB, as evidenced by the observed chemical shift pattern on the selenonium salt hydrogens. However, we were unable to obtain reliable results from 77Se NMR. While a very small shift was observed, prolonged exposure to benzyl bromide resulted in the decomposition of the selenonium salt.

Reviewer 4 Report

This manuscript entitled “Selenonium Salt as a Catalyst for Nucleophilic Substitution Reactions in Water: Synthesis of Thiocyanides and Selenocyanates” by Alix Y. Bastidas Ángel et. al. reports on the synthesis of organothiocyanates and organoselenocyanates by treating KChCN (Ch = S or Se) and selenonium salt C7 in H2O/DMC (20/1) solvent system.

             This study shows the interesting results based on the perspective of development of efficient methodology for synthesizing organothiocyanates and organoselenocyanates. Most of compounds were well synthesized in modest to good yields (4197%) in optimized reaction conditions. Furthermore, this study presents chalcogen bond interactions between a selenonium salt catalyst and an organic substrate by detecting the change of 1H NMR chemical shift of benzyl bromide in a solution of C7 in CDCl3. It was meaningful to show evidence for the important role of selenonium salt C7 in the designed reactions.

 So, publication of this manuscript is recommended after proper revisions. Here are some concerns about the present manuscript which need to be addressed by the authors before publication. 

  The abbreviation of ‘Nuclear Magnetic Resonance’ should have consistency. For example, there are ‘RMN’ expressions in the section of each compound’s spectral data despite ‘NMR’ expressions in the text. Please check the expressions in the whole manuscript.

The authors need to briefly mention about attempt of using other selenonium salts besides C7, if possible.

 In the title, some words need to be revised as such: Thiocyanides and Selenocyanates → Organothiocyanates and Organoselenocyanates

 In Table 3, # has to be substituted with ‘Entry’. 

- Check the numbering of headlines, spelling and word format throughly. There are some mistakes!          

Author Response

We want to thank the reviewer for the suggestions and comments. We have conducted a thorough review of the manuscript to correct any typos and incorrect nomenclature.

Currently, we have only tested the catalytic activity of one selenonium salt. However, we plan to expand our panel of catalysts to investigate how we can enhance catalytic performance by modulating the electronic environment on the selenium center.